

# Effects of Earth's magnetic field variation on high frequency wave propagation in the ionosphere

Mariano Fagre[1,2], Bruno S. Zossi[3,4], Erdal Yiğit[5], Hagay Amit[6], and Ana G. Elias[3,4]

[1]Consejo Nacional de Investigaciones Científicas y Técnicas, CONICET, Argentina.
[2]Laboratorio de Telecomunicaciones, Departamento de Electricidad, Electrónica y Computación, Facultad de Ciencias Exactas y Tecnología, Universidad Nacional de Tucuman, Argentina.
[3]Laboratorio de Física de la Atmosfera, Departamento de Física, Facultad de Ciencias Exactas y Tecnología, Universidad Nacional de Tucuman, Argentina.
[4]INFINOA (CONICET-UNT), Tucuman, Argentina.
[5]Space Weather Laboratory, Department of Physics and Astronomy, George Mason University, USA.
[6]CNRS, Université de Nantes, Nantes Atlantiques Universités, Laboratoire de Planétologie et de Géodynamique, Nantes, France.

*Correspondence to*: Mariano Fagre (mfagre@herrera.unt.edu.ar)

**Abstract.** The ionosphere is an anisotropic, dispersive medium for the propagation of radio frequency electromagnetic waves due to the presence of the Earth's intrinsic magnetic field and free charges. The detailed physics of electromagnetic wave propagation through a plasma is more complex when it is embedded in a magnetic field. In particular, the ground range of waves reflecting in the ionosphere presents detectable magnetic field effects. Earth's magnetic field varies greatly, with the most drastic scenario being a polarity reversal. Here the spatial variability of the ground range is analyzed using numerical ray tracing under possible reversal scenarios. Pattern changes of the "spitze", a cusp in the ray path closely related to the geomagnetic field, are also assessed. The ground range increases with magnetic field intensity and ray alignment with the field direction. For the present field, which is almost axial dipolar, this happens for Northward propagation at the magnetic equator, peaking in Indonesia where the intensity is least weak along the equator. A similar situation occurs for a prevailing equatorial dipole with Eastward ray paths at the corresponding magnetic equator that here runs almost perpendicular to the geographic equator. Larger spitze angles occur for smaller magnetic inclinations, and higher intensities. This is clearly observed for the present field and the dipole rotation scenario along the corresponding magnetic equators. For less dipolar configurations the ground range and spitze spatial variabilities become smaller scale. Overall, studying ionospheric dynamics during a reversal may highlight possible effects of dipole decrease which is currently ongoing.

## 1 Introduction

Radio frequency electromagnetic waves between 3 and 30 MHz, designated as high frequency (HF) waves by the International Telecommunication Union (ITU), are used in long-distance communications and detection, and have been of interest since the 1920's from a geophysical point of view as well as for practical reasons. Since the advent of telecommunication systems it has been a challenge to establish radio links as well as exact positions with radar systems using the ionosphere as a reflector due to the theoretical complexity of electromagnetic wave propagation through the ionospheric plasma that is embedded in the Earth's magnetic field. The ray tracing technique is commonly employed to solve this problem and to estimate the ray path



between the transmitter and a long-range target. Different methods have been developed, motivated by the appearance and fast progress of digital computers (Kelso, 1968; Settimi & Bianchi, 2014). The earliest research on methods of ionospheric ray tracing was stimulated by Haselgrove (1955) and Haselgrove and Haselgrove (1960). For a brief and interesting review see Bennett et al. (2004).

Ray tracing is a powerful and useful tool that allows determining the exact path of radio waves given a precise knowledge of
the ionospheric electron density profile along the propagation path, which is usually obtained from measurements or ionospheric modeling. Ray tracing can be assessed analytically or numerically to obtain the desired outputs, which are mainly the ground range, reflection height, phase and group path distances. The ground range is the measure of the distance along the surface of Earth from its origin (a transmitter) to the point where it again reaches Earth's surface (a receiver or a target); the reflection height is the true vertical distance of the point in the ray path where the ray is reflected by the ionosphere; the group-
path distance is the signal transmission time multiplied by free space light velocity c; and the phase-path distance is the wave-front travel time multiplied by c. A schematic geometry of a ray path and the above mentioned outputs is illustrated in Figure 1. For an analytical assessment, approximations must be made in order to be able to integrate and obtain explicit equations for the desired outputs. Therefore, only simple ionospheric and terrestrial magnetic field models can be considered. Alternatively, for a numerical assessment, numerical integration of the equations of electromagnetic wave propagation in a medium is
performed at each point and thus the ray tracing can be estimated using any ionospheric model, including the realistic terrestrial magnetic field and absorption effects. One numerical ray tracing technique that has been widely used is that of Jones and Stephenson (1975) who calculated ray paths and associated quantities in three-dimensional space using the spherical-polar coordinate system.

The presence of an ambient magnetic field in the ionosphere increases the complexity of plasma dynamics and electromagnetic
wave propagation. In addition, during disturbed magnetic conditions, variations of Earth's magnetic field have a great impact on ionospheric and thermospheric dynamics (Yiğit et al., 2016). The geometry of the magnetic field can lead to anisotropic effects in atmospheric gravity wave propagation and dissipation (Medvedev et al., 2017). These conditions indeed apply to Earth's ionosphere, which is a partially ionized plasma embedded in an intrinsic magnetic field, where many types of plasma waves can be triggered. Therefore, we focus our attention on magnetic field-induced effects on wave propagation that reflects
in the ionosphere, like effects on ground range, which are readily detectable and depend not only on the field intensity, but also on the angle between the ray and field vector.

Linked to ray paths through the ionosphere is the "spitze" phenomenon (Poeverlein, 1948), which occurs for the ordinary ray when the propagation is in the magnetic meridian for incident angles in the range between zero and a critical value that depends exclusively on the geomagnetic field intensity and inclination values (Davies, 1965; Budden, 1961). This critical angle, called
spitze angle, is in general useful for ionospheric modification by powerful radio waves experiments, and specifically ionospheric heating (Pedersen et al., 2003; Honary et al., 2011; Liu et al., 2018). According to Isham et al. (2005), "the maximum topside backscattered power occurs neither at the Spitze angle nor at field-aligned, but somewhere between." So, in some way, this angle value enters the equations that govern ionospheric changes produced by an intense radio wave.



Several works analyzed ray tracing in the ionosphere considering the geomagnetic field effect. Dao et al. (2016) found that
Earth's magnetic field effects can limit the accuracy of the mirror model (MM) used for geolocation. Tsai et al. (2010)
presented a ray tracing algorithm assuming an Earth-centered magnetic dipole. They obtained different values for the ground
range and the ray paths with vs. without their modeled geomagnetic field. These differences are less than 5%.

Paleomagnetic measurements show that Earth's magnetic field varies greatly, with the most dramatic case being polarity
reversals that take place on average every ~200 000 years (Jacobs, 1994; Glassmeier et al., 2009). However, the time period
of same polarity between reversals (termed chron) can be highly variable: tens thousands to tens millions of years (e.g. Olson
& Amit, 2015). The duration of a polarity reversal is a few thousand years (~1000-8000 years) (Clement, 2004). During this
period the field magnitude at the Earth surface may diminish to about 10% of its normal value. The last reversal occurred about
780,000 years ago (Jacobs, 1994).

The present field can be approximated by a geocentric magnetic dipole with its axis tilted about 11° with respect to Earth
rotational axis. This dipole accounts for ~80% of the magnetic field power at Earth's surface (Merrill et al., 1998). The
remaining ~20% is made up by non-dipolar components of the field.

Although the understanding of geomagnetic reversals has improved considerably over the years with paleomagnetic data
acquisitions and numerical geodynamo simulations, its properties, which involve the dominance of a dipolar or multi-polar
configuration during a polarity transition, are still under debate. Obviously the axial dipole vanishes during a reversal when
the dipole axis crosses the equator. Amit et al. (2010) summarized several reversal scenarios with two extremes for the dipolar
component: a dipole collapse and a dipole rotation from one hemisphere to the other.  In the latter case, only the axial dipole
would vanish by transferring its energy to the equatorial dipole components. Regarding the non-dipolar field, Amit et al. (2010)
considered three main possibilities: (1) decrease and recovery in phase with the dipole collapse, (2) remains unchanged, or (3)
grows throughout the reversal possibly due to energy transfer from the dipole (Amit & Olson 2010; Huguet & Amit 2012), or
dynamo configurations favoring the generation of a non-dipolar field.

In the present work the global spatial variability of the ground range and the spitze critical angle are determined for different
geomagnetic field configurations during reversals. These scenarios were recently considered to study the ionospheric Hall and
Pedersen conductances (Zossi et al., 2018) as well as the polar caps (Zossi et al., 2019), both during reversals.

Our models are a first approximation to the ground range variations due to different geomagnetic field configurations,
considering all other variables constant, even those processes affected by variations of the magnetic field itself. For example,
the equatorial ionization anomaly, which induces equatorial horizontal gradients in the electron density, may directly affect
ray tracing results. Even under geomagnetic quiet conditions, the terrestrial ionosphere is far from uniform. During the daytime
it strongly depends on solar zenith angle, which produces a latitudinal variation with greatest ionization at lower latitudes.
Earth's magnetic field, which regulates the transport processes of ions and electrons, can significantly influence the
morphology of the ionosphere. For example, ambipolar diffusion is an important transport process in the F2 layer. In general,
the thermosphere-ionosphere is continuously perturbed by a broad spectrum of upward propagating atmospheric waves (Yiğit
& Medvedev, 2017) as well as during transient lower atmospheric events, such as sudden stratospheric warmings. In the latter





case, for example, the equatorial ionization anomaly is modulated (Azeem et al., 2015) as a response to the changes in the electrodynamics.

Variations in Earth's magnetic field configuration (strength and morphology) can impact HF wave propagation, thermosphere-ionosphere dynamics, and manifestation of space weather. Inclusion of the geomagnetic field in ionospheric refractive index and maximum usable frequency calculations shows that the M(3000)F2 parameter increases, and hence hmF2, estimated using existing methods involving no magnetic field for M(3000)F2 scaling, would not capture this affect (Elias et al., 2017). With different magnetic field scenarios, the equatorial ionization anomaly is expected to follow the new "magnetic equators". In

particular, propagation of radio waves at high-latitudes is sometimes very different from propagation at middle and low latitudes, mainly due to the penetration of solar and magnetospheric particles that may create irregularities with scale sizes from meters to kilometers, greatly affecting HF ray paths. The greatest effects occur during geomagnetic storms and substorms since the high latitude ionosphere is the most affected in the present Earth's field configuration. Under a different magnetic field, such as the reversal scenarios considered here, polar caps and auroral zones would probably be completely different than

their current structure, which is then likely to change the nature of ion-neutral coupling at high-latitudes as well as the nature of the extension of storm-time effects to low-latitude regions. According to Zossi et al. (2019) an axial dipole collapse, for example, gives a pair of polar caps both at mid-latitudes of the southern hemisphere, while in a dipole rotation scenario the polar caps reside at the equator. If reversals occur due to an energy cascade from the dipole to higher degrees, more than two polar caps at various latitudes of both hemispheres prevail (Zossi et al., 2019).

Although there are several works which analyzed ray tracing results considering the geomagnetic field effect (Millington, 1951; Kelso, 1968; Rao, 1969; Bennett et al., 1991; Tsai et al., 2010; Dao et al., 2016; to mention a few) the consequences of the field's secular variation with a paleomagnetic perspective has not been studied yet. This is an interesting topic from a geophysical point of view and also for its potential applications in communications and radar systems. Although a reversal is foreseeable only in a very distant future, studying ionospheric dynamics during a reversal may highlight possible effects of

dipole decrease which is already ongoing at present (e.g. Gubbins, 1987; Olson & Amit, 2006; Huguet et al., 2018). It is thus necessary to systematically study various aspects of the geomagnetic field variations and their associated impact on radio wave propagation in the ionosphere.

In section 2 we outline the theoretical background of our study, followed by the methodology in section 3, which includes the description of the different possible reversal scenarios here considered in subsection 3.1, the ray tracing approach to determine

the ground range for HF propagation under the different field geometries in subsection 3.2, and the calculation set-ups for the ray tracing program in subsection 3.3. Results are presented in section 4 followed by discussion in section 5 and conclusions in section 6.





## 2 Theory

The Earth's magnetic field turns the ionosphere to an anisotropic medium, producing three main effects on the refraction of an incident electromagnetic wave: (i) double refraction, (ii) the direction of energy flow differs from that of the wave propagation, and (iii) the refractive index depends on the angle of refraction hindering the solution of Snell's law directly. Regarding the first effect, the ionosphere, as any magnetized plasma, becomes a doubly refracting medium decomposing an incident linearly polarized wave into two modes of propagation for which the terms "ordinary" and "extraordinary" are taken

from crystal optics, with subscripts "o" and "x" denoting each mode respectively. These are two elliptically polarized waves of opposite rotational sense: right-hand and left-hand polarization in the cases of the o- and x-mode, respectively. We will consider the ordinary ray mode, which is visible most of the time. The second effect turns asymmetrical the ray path of an oblique propagating electromagnetic signal with respect to the reflection point, and deviates the path laterally out of the plane of incidence. Regarding the third, the refractive index n, assuming a cold magnetoplasma where only electrons need to be

taken into account (valid approximation for the propagation of HF signals in the ionosphere), is given by the Appleton-Hartree equation (Ratcliffe, 1962) where only electrons need to be taken into account, that is,

$$n = \sqrt{1 - \frac{2X(1-X)}{2(1-X)-Y_T^2 \pm \sqrt{Y_T^2+4(1-X)^2 Y_L^2}}} \qquad (1)$$

with

$$X = \frac{f_o^2}{f^2} = \frac{Ne^2}{m(2\pi f)^2} \qquad (2)$$

$$Y_T = Y\sin\Theta = \frac{eB}{m(2\pi f)}\sin\Theta \qquad (3)$$

$$Y_L = Y\cos\Theta = \frac{eB}{m(2\pi f)}\cos\Theta \qquad (4)$$

$$Y = \frac{eB}{m(2\pi f)} \qquad (5)$$

where $f_o$ is the plasma frequency, f the incident electromagnetic wave frequency, N the electron number density, e the electron charge, m the electron mass, B the magnetic field intensity, T stands for transverse and L for longitudinal, and the angle Θ

corresponds to the angle between the direction of the wave propagation and the magnetic field vector. The upper sign in the denominator of Equation (1) refers to the ordinary component and the lower sign to the extraordinary.

Equation (1) deviates from the solution of Snell's law for the ray path of a given electromagnetic wave. This problem can be solved using ray tracing based on Hamilton's equations, which is used in the present work and is described in subsection 3.2.

Two limits of Equation (1) are worth noting. When the wave is perfectly field-aligned Θ=0°, $Y_T = 0$ and $Y_L = Y$, giving for

the ordinary wave

$$n(\Theta = 0°) = \sqrt{1 - \frac{X}{1+Y}} \qquad (6)$$

This means that for higher field intensity n is closer to the free space value (n=1). In contrast, when the wave is perpendicular to the field lines Θ=90°, $Y_T = Y$ and $Y_L = 0$, giving for the ordinary wave



$$n(\Theta = 90°) = \sqrt{1-X} \hspace{4cm} (7)$$

that corresponds to n for a non-magnetized ionosphere.

In general, the angle $\Theta$ is given by

cos($\Theta$) = cos(I) cos(α) [cos(D) cos(γ) + sin(D) sin(γ)] + sin(I) sin(α)          (8)

where D and I are the geomagnetic field declination and inclination, respectively, α the elevation angle of the electromagnetic signal emitted by the transmitter, and γ the direction of the ray path, that is 0° and 90° for Northward and Eastward propagation

respectively.

When the propagation is in the magnetic meridian and the incidence angle lies between zero and a critical angle, $\phi_c$, the ordinary wave ray path never becomes horizontal. The refractive index drops suddenly to the reflection condition producing a discontinuity or cusp in the ray path called "spitze", a term adapted from German (meaning "pointed" or "sharp") by Poeverlein (1948) who discovered this phenomenon (Davies, 1965; Budden, 1961; Huang & Reinisch, 2006). Figure 2 shows a schematic

illustration of the spitze obtained from the original paper by Poeverlein (1948). $\phi_c$, also called sptize angle, depends exclusively on the geomagnetic field intensity and inclination values (Poeverlein, 1948; Davies, 1965; Budden, 1961) and is given by

$$\Phi_c = \sqrt{\frac{Y}{1+Y}} \, cos(I) \hspace{4cm} (9)$$

This angle marks the end of the spitze region. From this equation $\Phi_c$ can be estimated for any magnetic field configuration. For the present field, considering the almost axial dipolar configuration, the greatest $\Phi_c$ is expected along the magnetic equator.

**3 Methodology**

**3.1 Earth's magnetic field configurations**

The International Geomagnetic Reference Field 12th Generation (IGRF-12) (Thébault et al., 2015) was used to specify the pre-reversal magnetic field **B** for all scenarios. **B** is given in terms of the internal magnetic scalar potential V by **B**=-∇V, which is expanded by the series

$$V(r,\theta,\varphi,t) = a \sum_{n=1}^{13} \sum_{m=0}^{\ell} \left(\frac{a}{r}\right)^{\ell+1} \{[g_\ell^m(t)cos(m\varphi) + h_\ell^m(t)sin(m\varphi)]P_\ell^m(cos\theta)\}, \hspace{1cm} (10)$$

where a = 6371.2 km is Earth's mean reference spherical radius, r the radial distance from the center of the Earth, θ the geocentric co-latitude, φ the East longitude, $P_n^m(cos\theta)$ the Schmidt quasi-normalized associated Legendre functions of degree $\ell$ and order m, and $g_n^m$ and $h_n^m$ the Gauss coefficients which are functions of time t (e.g. Merrill et al., 1998).

The reversing field was modeled first by gradually decreasing the coefficients of the dipolar components, that is those

corresponding to $\ell$ =1 ($g_1^0$, $g_1^1$ and $h_1^1$), while keeping unchanged the quadrupolar and octupolar coefficients. The three other end member scenarios considered for the reversal are: an axial dipole collapse where the axial dipolar component is set to zero while maintaining the equatorial dipole components as well as higher degrees unchanged (that is only setting $g_1^0$=0), a dipole rotation where the power of the axial dipole component is transferred to the equatorial dipole components proportional to their pre-reversal powers (that is setting $g_1^0$=0 and increasing $g_1^1$ and $h_1^1$), and a third scenario consisting of an energy cascade





where the power of the dipolar components is transferred to the quadrupolar and octupolar components also proportional to the pre-reversal power of each degree and order (that is setting $g_1^0$, $g_1^1$ and $h_1^1$ to zero and increasing the next 12 Gauss coefficients which correspond to the 5 quadrupolar and the 7 octupolar terms). For the last two scenarios, a constant total magnetic power on the core-mantle boundary calculated based on the Mauersberger–Lowes spectrum (Lowes, 1974) was considered, given by

$$R = \sum_\ell R_\ell = \sum_\ell (\ell + 1) \left(\frac{a}{c_r}\right)^{2\ell+4} \sum_m [(g_\ell^m)^2 + (h_\ell^m)^2] \tag{11}$$

where $c_r$=3480 km is the radius of the core. The configuration of the remaining components in each case (the equatorial dipoles in the first, and the quadrupolar and octupolar components in the second) was maintained by keeping the respective proportions constant. That is, given $g_1^0$, $g_1^1$ and $h_1^1$ for present conditions, the dipole rotation reversal scenario (denoted by asterisk superscript) consists of $g_1^{*0}$ =0 and $g_1^{*1}$ and $h_1^{*1}$ given by

$$\frac{2 \left(\frac{a}{c_r}\right)^6 (g_1^{*1})^2}{R_1} = \frac{2 \left(\frac{a}{c_r}\right)^6 (g_1^1)^2}{R_1 - 2 \left(\frac{a}{c_r}\right)^6 (g_1^0)^2}$$

$$\tag{12}$$

$$\frac{2 \left(\frac{a}{c_r}\right)^6 (h_1^{*1})^2}{R_1} = \frac{2 \left(\frac{a}{c_r}\right)^6 (h_1^1)^2}{R_1 - 2 \left(\frac{a}{c_r}\right)^6 (g_1^0)^2}$$

where $R_1 = R_1^*$, that is $(g_1^0)^2+(g_1^1)^2+(h_1^1)^2=(g_1^{*1})^2+(h_1^{*1})^2$. For the energy cascade scenario, the transfer to the quadrupole and octupole coefficients considering $g_1^{*0} = g_1^{*1} = h_1^{*1} = 0$ is given by

$$\frac{(\ell + 1) \left(\frac{a}{c_r}\right)^{2\ell+4} (g_\ell^{*m})^2}{R} = \frac{(\ell + 1) \left(\frac{a}{c_r}\right)^{2\ell+4} (g_\ell^m)^2}{R_2 + R_3}$$

$$\tag{13}$$

$$\frac{(\ell + 1) \left(\frac{a}{c_r}\right)^{2\ell+4} (h_\ell^{*m})^2}{R} = \frac{(\ell + 1) \left(\frac{a}{c_r}\right)^{2\ell+4} (h_\ell^m)^2}{R_2 + R_3}$$

Here $R=R_2^*+R_3^*$, since $R_1^*=0$.

## 3.2 HF signal ray tracing procedure

Various numerical ray tracing programs have been developed. Among them, Azzarone et al. (2012) developed a software that is freely available and allows ionospheric ray tracing in a geocentric spherical coordinate system, taking into account a dipolar geomagnetic field. However, since we consider multi harmonic scenarios, the 3D ray tracing original code developed in the work by Jones and Stephenson (1975) was adjusted to include the IGRF-12 model and the configurations for the transitional



magnetic field. This ray tracing is based on Hamilton's equations of geometrical optics given by Haselgrove (1955) in spherical coordinates.

The Hamiltonian used here is given by

$$H_{(r,\theta,\varphi,k_r,k_\theta,k_\varphi)} = \frac{1}{2} * \Re\left[\frac{c^2}{\omega^2}\left(k_r^2 + k_\theta^2 + k_\varphi^2\right) - n^2\right] \tag{14}$$

where $k_r$, $k_\theta$ and $k_\varphi$ are the spherical components of the HF wavenumber vector of angular frequency $\omega=2\pi f$.

From Hamilton's equations of motion $-dp_i/dt=\partial H/\partial q_i$ and $dq_i/dt=\partial H/\partial p_i$. In this case the generalized coordinate $q_i$ corresponds to r, θ and φ, the generalized momentum $p_i$ corresponds to $k_r$, $k_\theta$ and $k_\varphi$, and instead of t, ct is used, so that d(ct)=dτ, where τ is the HF wave group path. The differential dτ is connected with the element of arc length ds along the ray path through a point of coordinates (r, θ, φ) by the relation dτ = n ds. In this way, Hamilton's equations, which consist of a set of six partial differential equations, are

$$\frac{dr}{d\tau} = \frac{\partial H}{\partial k_r} \tag{15}$$

$$\frac{d\theta}{d\tau} = \frac{1}{r}\frac{\partial H}{\partial k_\theta} \tag{16}$$

$$\frac{d\varphi}{d\tau} = \frac{1}{rsin(\theta)}\frac{\partial H}{\partial k_\varphi} \tag{17}$$

$$\frac{dk_r}{d\tau} = \frac{-\partial H}{\partial r} + k_\theta\frac{d\theta}{d\tau} + k_\varphi sin(\theta)\frac{d\varphi}{d\tau} \tag{18}$$

$$\frac{dk_\theta}{d\tau} = \frac{1}{r}\left(\frac{-\partial H}{\partial \theta} - k_\theta\frac{dr}{d\tau} + k_\varphi rcos(\theta)\frac{d\varphi}{d\tau}\right) \tag{19}$$

$$\frac{dk_\theta}{d\tau} = \frac{1}{rsin(\varphi)}\left(\frac{-\partial H}{\partial \varphi} - k_\varphi sin(\theta)\frac{dr}{d\tau} - k_\varphi rcos(\theta)\frac{d\theta}{d\tau}\right) \tag{20}$$

The software to obtain the ray path and the integration algorithm are from the Fortran code by Jones and Stephenson (1975), including the improvements and corrections made by Azzarone et al. (2012).

### 3.3 Calculation setup


The global spatial structure of ground range for an oblique propagation and of the spitze critical angle were assessed on a grid with 5° latitude and 10° longitude resolution. In the case of the ground range, in order to analyze changes due only to Earth's magnetic field origin, a horizontally uniform ionosphere was considered. That is, a single electron density height profile was used for the whole grid, consisting in this case in a β-Chapman layer. The plasma frequency $f_o$ is then given by

$$f_o^2 = foF2^2 exp(1 - z - e^{-z}) \tag{21}$$

where

$$z = \frac{h - hmF2}{H} \tag{22}$$



foF2 and hmF2 are the critical frequency and the peak height of the F2 layer respectively, which are obtained from the
International Reference Ionosphere (IRI) model for 12 LT, a quiet day in April during solar minimum conditions (April 2,
2008), that is foF2=8 MHz and hmF2=300 km. An isothermal ionosphere was considered with the typical 60 km value for the
neutral scale height H, considering atomic oxygen as the main ionizable neutral component at the F2 region.

Northward and Eastward wave propagation directions were considered with a fixed elevation angle, α = 20°, and a single
frequency of 15 MHz, which are typical mean values for over the horizon radars, OTHR, which use Earth's ionosphere as a
mirror to illuminate targets beyond the line-of-sight horizon.

## 4 Results

Figure 3 shows the field intensity B (left column) and cos(I) (right column) obtained from the IGRF-12 model for three
different field scenarios: present field conditions (first row), 50% (second row) and 90% (third row) decrease of dipolar
components. Figure 4 presents B and cos(I) for three additional reversal scenarios: axial dipole collapse (first row); dipole
rotation where axial dipole energy is transferred to the equatorial dipole (second row); and energy cascade where the dipolar
energy is transferred to the quadrupolar and octupolar terms (third row). Figures 5 and 6 present the corresponding ground
range distributions for Northward (left column) and Eastward (right column) wave directions.

As the dipolar component of the geomagnetic field decreases, not only a global intensity decrease is noticed (see different
scales in Figure 3), but also the strongest intensity moves from the geographic poles and the four high-latitude intense flux
patches (e.g. Jackson et al., 2000) to other locations in East Asia and South Atlantic. In addition, the inclination I and
consequently cos(I) become less zonal, in particular in the South Atlantic. In the other three reversal scenarios the axial dipole
component is zero hence the intensity and inclination become even more meridional (Figure 4).

The ray path, and hence the resulting ground range, are uniquely determined by the refractive index n. This index depends in
a non-trivial way on the field intensity B and the ray-field angle Θ (Equation (1)). This angle depends on I and D (Equation
(8)). Therefore, the relationship between the field intensity and inclination (Figures 3 and 4) and the resulting ground ranges
(Figures 5 and 6) is not straightforward. According to Equation (1) n is closer to its free space value for increasing B and/or
for lower Θ, i.e. for ray paths aligned with the field (see Equation (6)). Indeed, the largest ground range values observed in
Figures 5 and 6 occur, when the ray becomes horizontal and is reflected at higher altitudes, for the lowest Θ values, i.e. for
cos(Θ) closer to 1. This is evident by comparison with Figure 7 which presents the global pattern of cos(Θ) for three field
configurations for Northward and Eastward propagations.

To describe in detail how the field configuration and its interaction with the wave propagation determine the ground range,
consider the relations among the present day field intensity (Figure 3a), the resulting ground ranges for Northward (Figure 5a)
and Eastward (Figure 5b) propagations, and the respective field-wave angles (Figure 7a and b). The field is most aligned with
the Northward wave in the magnetic equator (Figure 7a), but in this region the intensity of the axial dipole dominated present
field is lowest (Figure 3a). The resulting ground range is largest in Indonesia (Figure 5a) where the intensity is least low in the





equatorial belt (Figure 3a). In the case of the Eastward wave the field is most aligned in the Pacific (Figure 7b) where the intensity is also high (Figure 3b), hence the ground range is largest there (Figure 5b).

In the dipole rotation scenario, the field is most aligned with the Northward wave at vast regions of high latitudes of the northern hemisphere and at a small region near the South Pole (Figure 7c). However, the intensity is strongest at lower latitudes

in the longitudes of East Asia and the Americas (Figure 4c). The resulting ground range is largest in the overlapping regions of high latitudes at these two longitudes of equatorial pole and anti-pole, most notably near the South Pole (Figure 6c). For the Eastward wave the field alignment approximates a spherical harmonic degree and order 2 pattern with maxima all along the longitudes of Africa and the Pacific (Figure 7d) which is well correlated with the ground range (Figure 6d) despite the apparent anti-correlation with the intensity (Figure 4c).

Finally, in the energy cascade scenario, for the Northward wave the ground range is largest in an east-west strip in the south Atlantic (Figure 6e) where both the intensity and the field alignment are large (Figures 4e and 7e). Similarly, for the Eastward wave the ground range is largest in a north-south strip in the Indian Ocean (Figure 6f) where both the intensity and the field alignment exhibit high values (Figures 4e and 7f). Note the resemblance between the ground ranges in the scenarios of energy cascade and axial dipole collapse (compare Figures 6e and f with Figures 6a and b) due to the similar field morphologies of

these cases, in analogy to the resemblance of the polar caps patterns associated with these two scenarios (Zossi et al., 2019).

Overall, a rather good agreement between the wave to field lines alignment and the ground range in the different magnetic field scenarios can be noticed (compare Figures 7a and b with Figures 5a and b; Figures 7c and d with Figures 6c and d; Figures 7e and f with Figures 6e and f). However, due to the additional dependence on the field intensity, some discrepancies exist. For example, in the energy cascade scenario the ray-field alignments are characterized by numerous small scale features

(Figures 7e and f) whereas the maximal ground ranges are localized at single locations where overlaps with high intensity structures prevail (Figures 6e and f).

Figure 8 presents the spitze angle for the six field configurations. As expected from Equation (9), the spitze angle closely follows the cos(I) pattern (right panels of Figures 3 and 4). Here again, the most noticeable variation appears in the spatial variability. The greatest spitze angle values are 15.5° for the present field (Figure 8a) and 15.3° for the dipole rotation scenario

(Figure 8e). The lowest peak value is 9.4° for the 90% decrease of the axial dipolar component. Overall, large/small spitze peak values are associated with large-scale/small-scale inclination patterns, respectively.

## 5 Discussion

In the absence of a magnetic field, for the ray tracing conditions considered here (f=15 MHz and 20° elevation angle), we would obtain a uniform ground range distance of 1390 km for both propagation directions (Northward and Eastward) for the

whole globe. With the present magnetic field, the ground range is no longer uniform (Figures 5a and b), varying between ~1390 and ~1420 km for Northward propagation and between ~1390 and ~1400 km for Eastward propagation. These extreme values differ very little among the considered field scenarios, by at most 2%. The main variation in the global pattern is related




to the cos(Θ) value, which in turn depends on the ray direction, and the field inclination and declination global distributions. In addition, the ground range depends on the field intensity in a non-trivial way.

Inside Earth's outer core, the alignment of the fluid flow and the magnetic field determines the efficiency of induction. In particular, at the top of the core horizontal flow that is parallel to radial field isolines produces zero advective secular variation, which is a severe inherent problem for inferring the flow from geomagnetic observations (e.g. Backus & Bullard, 1968). Non-linear magnetohydrodynamic effects tend to align the flow and the field hence to minimize their interaction (Aubert, 2005; Cao et al., 2018). At the top of the core, numerical dynamos exhibit large alignment at high-latitude intense flux patches which

are correlated with axial columnar cyclones that maintain these features, whereas low-latitude drifting field structures are advected by a flow that is nearly perpendicular to these patches (Finlay & Amit, 2011). This bimodal behavior gives an intermediate value for the globally averaged alignment (Finlay & Amit, 2011; Peña et al., 2016). Overall, understanding the flow-field alignment in the core is crucial for properly inferring core dynamics, just as accounting for wave-field alignment in the ionosphere reveals the spatial variability of ground range.

The greatest difference between ground range values for the present field and the other scenarios considered here is around 30 km. Even though small in percentage terms, these values could matter for certain applications, such as higher order ionospheric effects for trans-ionospheric propagation that are increasingly relevant as precision requirements on GPS (Global Positioning System) data and products increase. In this respect, the ionosphere is a significant error source for Global Navigation Satellite Systems (GNSS) and GPS data. These errors may vary from a few centimeters to tens of meters. Most of it can be corrected

by combining signals at two frequencies. However, higher order errors remain, such as the second order ionospheric term in the refractive index formula which depends precisely on the geomagnetic field. These errors become some of the main limiting factors for applications where millimeter level accuracy is demanded (Petrie et al., 2011; Hoque & Jakowski, 2008; Banville et al., 2017; Hadas et al., 2017). The dipole model, which is a simpler representation of Earth's geomagnetic field than the true multi harmonic field, is sometimes used. However, even though the resulting coordinate differences are not large, considering

the true field is crucial (Petrie et al., 2011), since the difference in the second order error between the dipole and IGRF model can be up to 60%, mainly in the South Atlantic Anomaly (Hernandez-Pajares et al., 2007).

The ground range increase for an increasing field intensity can also be deduced from the dielectric constant ε, which for the ionosphere, treated as a cold magnetized plasma, is a tensor. The square refractive index is equal to ε, and this in turn is equal to $1+4\pi i\sigma/\omega$. σ is the alternate current induced by the electromagnetic wave of frequency ω in the cold ionosphere, and

decreases with an increasing B, as is the case of the direct conductivity in a "warm" magnetized plasma analyzed in Zossi et al. (2018). In this way, for increasing B, ε tends to 1 which is the free space value, favoring greater ground ranges for frequencies which are reflected by the ionosphere.

**6 Conclusions**

We studied the variation of HF signal propagation in the ionosphere under various configurations of the internal core magnetic

field. For this, the current magnetic field is retrieved from the latest IGRF model to represent a pre-reversal state and then the





transitional field is modeled according to the following cases: 50% and 90% decrease of dipolar components; axial dipole collapse; dipole rotation where axial dipole energy is transferred to the equatorial dipole; and energy cascade where the dipolar energy is transferred to the quadrupolar and octupolar terms. The main findings of our study are:

1) For the present field configuration, which is dominantly axial dipolar, greater ground range values are obtained for Northward ray paths around the magnetic equator. This happens because the ground range of an oblique HF signal ray path is greater for greater alignment with the field lines direction. A similar situation occurs for a prevailing equatorial dipole (dipole rotation scenario) with Eastward ray paths around the corresponding magnetic equators, which in this scenario run through the longitudes of Africa and the mid-Pacific Ocean.

2) When the wave propagates perpendicular to the magnetic equator, e.g. Eastward for the present field and Northward for the dipole rotation scenario, maximal ground ranges are non-trivially concentrated near the South Pole. In the first case this is due to greater alignment near the South Pole (Figure 7b). In the second case, the reason is the greater magnetic field intensity which almost doubles that of the North Pole.

3) For less dipolar configurations, the ground range spatial variability becomes less symmetric. For both wave directions in these scenarios localized ground range maxima appear near the South Atlantic Anomaly, a region where the present field is anomalously weak with large non-dipolar contributions (Terra-Nova et al., 2017) but the intensity of the transitional field at this region is no longer minimal.

4) The ground range enhancement with respect to the no-field situation is somewhat larger for Northward propagation, especially at the beginning of a reversal when the dipole is still strong (Figures 5a-d).

5) The spitze critical angle, which exists only in a magnetized plasma, has greatest values along the magnetic equator i.e. for zero field inclination, and for higher field intensity. This is clearly observed for the present field configuration case and, again, for the dipole rotation scenario. The spatial variability becomes smaller scale for the less dipolar configurations.

6) Large/small spitze peak values are associated with large-scale/small-scale inclination patterns, respectively. Non-dipolar configurations, such as the 90% dipolar decrease, the axial dipole collapse and the energy cascade scenarios, present smaller scale patterns. The lower field intensity results in lower $\sqrt{\frac{Y}{1+Y}}$ values, and consequently lower $\Phi_c$.

The ground range dependence on the geomagnetic field and the existence of the spitze phenomenon are just a couple of the many features of electromagnetic wave propagation in a magnetized plasma. As stated in the seminal review paper by Yeh and Liu (1972), "one of the most outstanding features of a plasma is the change of its electromagnetic properties when it is under the influence of an external steady magnetic field". This is precisely the case of the ionosphere embedded in Earth's magnetic field. Overall, unravelling the electromagnetic properties of the ionospheric plasma during a reversal may highlight possible effects of dipole decrease which is currently ongoing.




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

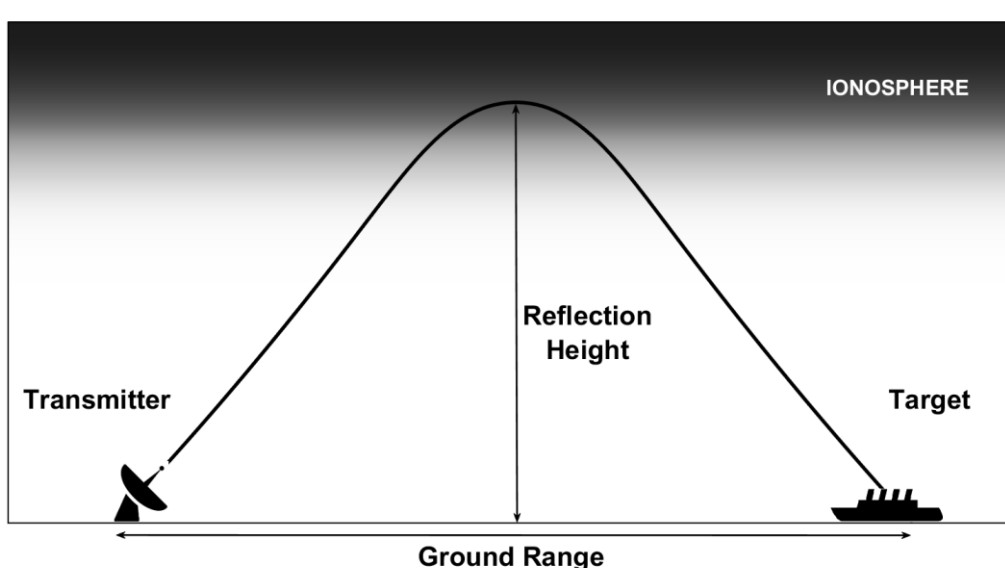

**Figure 1. Schematic illustration of the geometry of a ray path through the ionosphere.**




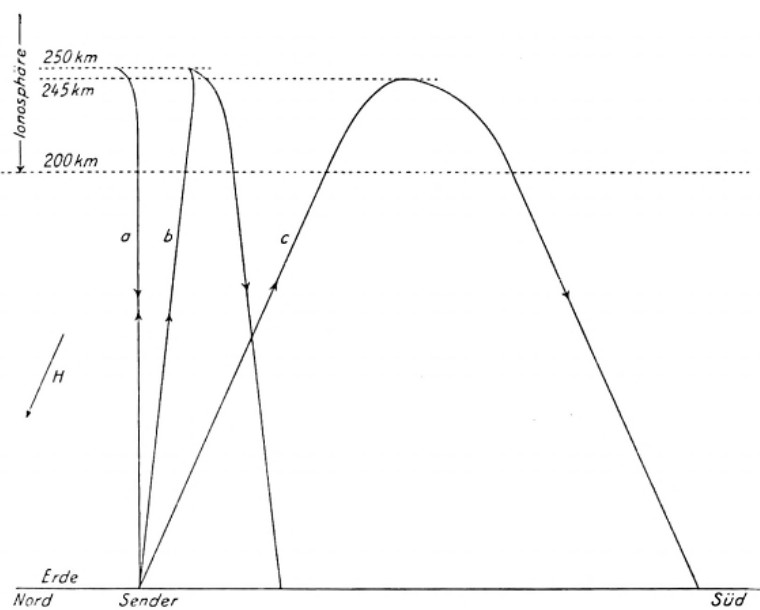

Abb. 3. Strahlwege des ordentlichen Strahls (Daten S. 187)

H = Richtung des Erdmagnetfelds

(a)


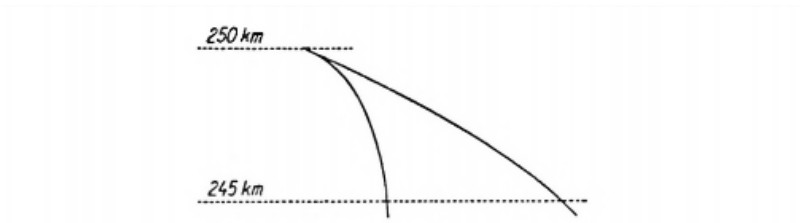

Zu Abb. 3. Vergrößerte Darstellung der Spitze von Strahlweg *b*

(b)

**Figure 2. Schematic illustration of the sptize shown in Figure 3 of the original paper by Poeverlein (1948). (a) Ordinary ray paths for increasing incident angle Φ: *a* corresponds to Φ=0, *b* to Φ<Φ$_c$, and *c* to Φ>Φ$_c$. Only rays *a* and *b* show a spitze. (b) Enlarged view**

**of the "spitze" of the ray path. (Permission to reproduce the illistration granted by Bayerische Akademie der Wissenschaften)**





**Figure 3. Intensity of Earth's magnetic field B [nT] and cos(I), where I is the inclination, obtained from IGRF-12 for: (a) and (b) present conditions, (c) and (d) 50% decrease of dipolar components, (e) and (f) 90% decrease of dipolar components. Note the different scales for intensity.**






(a)   (b)

(c)   (d)

(e)   (f)

**Figure 4.** As in Fig. 3 for (a) and (b) axial dipole collapse, (c) and (d) dipole rotation where axial dipole energy is transferred to the equatorial dipole, and (e) and (f) energy cascade where dipolar energy is transferred to the quadrupolar and octupolar terms. Note the different scales for intensity.








**Figure 5.** Ground range [km] for an HF wave of 15 MHz and a 20° elevation angle for: (a) and (b) the present Earth's magnetic field, (c) and (d) a 50% decrease of dipolar components, and (e) and (f) a 90% decrease of dipolar components. Left panels correspond to Northward and right panels to Eastward propagation. Note the different scales.







**Figure 6. As in Figure 5 for: (a) and (b) an axial dipole collapse, (c) and (d) a dipole rotation where axial dipole energy is transferred to the equatorial dipole, and (e) and (f) an energy cascade where dipolar energy is transferred to the quadrupolar and octupolar terms. Note the different scales.**




**Figure 7.** cos(Θ), where Θ is the angle between the incident HF wave at the base of the ionosphere and the Earth's magnetic field
direction for: (a) and (b) the present Earth's magnetic field, (c) and (d) a dipole rotation where axial dipole energy is transferred to
the equatorial dipole, and (e) and (f) an energy cascade where dipolar energy is transferred to the quadrupolar and octupolar terms.
Left panels correspond to Northward and right panels to Eastward propagation.




(a)                                    (d)

(b)                                    (e)

580          (c)                                    (f)

**Figure 8. Spitze angle for (a) the present Earth's magnetic field, (b) a 50% decrease of dipolar components, (c) a 90% decrease of dipolar components, (d) an axial dipole collapse, (e) a dipole rotation where axial dipole energy is transferred to the equatorial dipole, and (f) an energy cascade where dipolar energy is transferred to the quadrupolar and octupolar terms. Note the different scales.**
