# Peer review of "Effects of Earth's magnetic field variation on high frequency wave propagation in the ionosphere"

_Annales Geophysicae, 2019_

## Referee Comment (RC1) · Anonymous Referee #1 · 2 Apr 2019

**Review of the manuscript** entitled "Effects of Earth's magnetic field variation on high frequency wave propagation in the ionosphere" by Mariano Fagre et al., submitted for a possible publication in Annales Geophysicae [angeo-2019-27]

**General comment**

The manuscript describes influence of substantial changes of the Earth's magnetic field that might take place in future on propagation of high frequency (HF) radio waves in the ionosphere. The propagation and downward refraction of the radio waves in the ionosphere is studied by 3D ray tracing code. The authors mainly focus on the changes of ground range R for the waves transmitted at specific frequency and elevation angle, and partly also on the changes of the so-called Spitze angle.

The main problem of the presented study is that it does not take into account any variability of electron densities in the ionosphere, which has a major effect on HF wave propagation and hence on the ground range R, whereas the changes in magnetic field have only minor effect. The authors themselves found that changes in R owing to relatively drastic variations of the Earth's magnetic field (dipole collapse or reversal) are by at most 2% for globally constant electron distribution. I expect that such R changes are much lower than R changes owing to diurnal and seasonal variability of the ionosphere and also much lower than uncertainties in the calculated R owing to uncertainties in ionospheric model, ionospheric disturbances etc. I miss any comparison with these ionospheric variations and uncertainties. Moreover, it is reasonable to expect that such dramatic future changes of the Erath's magnetic field will be associated with global changes of electron densities as the authors also partly mention in the Introduction. Therefore, I consider the presented study a sophisticated workout on ray tracing code rather than a useful geophysical investigation.

I recommend the authors to mainly focus on the Spitze angle, and discussed this point more in detail. The Spitze angle only depends on the magnetic field. Thus, the calculated changes of Spitze angle due to magnetic field variations are meaningful, unlike the changes of R which dominantly depend on ionosopheric density and its variability. The minor changes of R due to magnetic field variation could be shortly mentioned in discussion for completeness.

I have also found several incorrect or confusing statements in the text (see the specific comments).

For all the reasons mentioned above, I cannot recommend the manuscript for publication in the present form. The manuscript requires substantial revision in my opinion.

**Specific comments**: (As I expect major revision and changes in the text, only major comments are listed. I do not provide formal or language comments at this stage.)

a) The second part of abstract is difficult to understand without reading the article

b) lines 44-46. Definitions of group and phase path using speed of light $c$ in vacuum are misleading. According to my knowledge and literature that I read the phase and group path lengths are related to distances traveled by phase and group velocities along the trajectory, respectively (group path length is simply the length of the trajectory). Anyway, I think that these terms are unnecessary for the purpose of this article and could be removed.

c) paragraph on lines 54-61. I think it could be removed as similar information is better described later, e.g, in the text starting in the end of line 100 and in the following paragraph

d) line 141, "*right-hand and left-hand polarization in the cases of the o- and x-mode, respectively*" That is incorrect. Ordinary mode is left-handed (L), whereas extraordinary mode is right-handed (R). The terms L-O and R-X modes are often used, instead of simply O and X modes.

e) Permittivity of vacuum is missing in equation (2)

e) lines, 157-158, remove

f) Equation (8), specify that $\theta$ changes along the ray path in the ionosphere. Using the initial values, it is only valid at the bottom or below the ionosphere.

g) Text related to equation (9), lines 171-179. It should be mentioned that Spitze trajectories for incidence angles between zero and critical angle $\Phi_c$, $0 < |\Phi| < \Phi_c$, are only formed for wave frequencies $f$<foF2.

h) I think there should be $\sin(\Phi_c)$ on the left hand side of equation (9),
see, e.g., Eliasson et al. (2015), J. Plasma Physics, vol. 81, 415810201,
doi:10.1017/S0022377814000968  or
Mjolhus (1990), Radio Sci. **25**(6), 1321–1339

i) Is really the same ionospheric profile used over the same globe? If yes, it makes no sense. See also the general comment

j) lines 268-269. The refractive index mainly depends on plasma density. Magnetic field **B** and angle betwenn **B** and **k** have relatively minor effect in the ionosphere.

k) line 280, least low in equatorial belt-> largest in equatorial belt
(the same in abstract)

l) lines 325-342, this is useless here and partly out of context; without discussing the dominant effect of electron densities it makes no sense.

m) lines 370-375. This text is suitable for Introduction rather than for conclusion.

---

## Author Comment (AC1) · 8 Apr 2019

**Answer to Reviewer #1:**
Thank you very much for your comments and corrections.
Following are our answer (in black) to each of your comments (in red).

**General comment**
The manuscript describes influence of substantial changes of the Earth's magnetic field that might take place in future on propagation of high frequency (HF) radio waves in the ionosphere. The propagation and downward refraction of the radio waves in the ionosphere is studied by 3D ray tracing code. The authors mainly focus on the changes of ground range R for the waves transmitted at specific frequency and elevation angle, and partly also on the changes of the so-called Spitze angle.
The main problem of the presented study is that it does not take into account any variability of electron densities in the ionosphere, which has a major effect on HF wave propagation and hence on the ground range R, whereas the changes in magnetic field have only minor effect. The authors themselves found that changes in R owing to relatively drastic variations of the Earth's magnetic field (dipole collapse or reversal) are by at most 2% for globally constant electron distribution. I expect that such R changes are much lower than R changes owing to diurnal and seasonal variability of the ionosphere and also much lower than uncertainties in the calculated R owing to uncertainties in ionospheric model, ionospheric disturbances etc. I miss any comparison with these ionospheric variations and uncertainties. Moreover, it is reasonable to expect that such dramatic future changes of the Erath's magnetic field will be associated with global changes of electron densities as the authors also partly mention in the Introduction. Therefore, I consider the presented study a sophisticated workout on ray tracing code rather than a useful geophysical investigation.

The main idea of this work is to isolate the field effect. Indeed natural electron density variations (daily, seasonal, decadal due to solar activity, etc.) have a stronger effect on R than the natural variation of the Earth's magnetic field, even during reversal scenarios – we will add to the text a thorough comparison between the effects of electron density and magnetic field morphology on the ground range. However, the effect of the magnetic field is important for two main reasons:
1. from a theoretical point of view.
2. electron density is characterized by large amplitude high frequency variations (days, annual, decadal), whereas the magnetic field is characterized by low amplitude lower frequency variations. Our study reveals the ground range variability over a timescale of several thousand years of a reversal.

Maybe a better way to present our results would be to show R differences between each scenario and the present one. Even though the absolute value of the differences will depend on the electron density, the sign (positive for an increase or negative for a decrease) will be the same in all cases.

Regarding the magnetic field effect on electron density, it affects mainly the F2 region. The lower layers of the ionosphere that is the E and F1 regions are mainly under a photo-chemical regime, so changes in the solar radiation and solar zenith angle are expected to be the dominant variation sources. In the case of the F2 layer, plasma diffusion becomes important and the magnetic field plays its role by reducing the effectiveness of diffusion due to ions and electrons

which are forced to diffuse along B at these heights. Changes in the magnetic field inclination and declination then can move up and down the F2 peak height affecting the ionization density. Energetic particle precipitation, which is also a source of ionization, would present stronger changes with the magnetic field variations. However, it is a source of transient character. Cnossen et al. (2011) estimated for a 25% reduction in the dipole moment of the Earth an increase in temperature, causing the thermosphere to expand and ionospheric layers to move upwards, since they tend to stay on constant-pressure surfaces. They found that the electron density is more affected at equinox with a ~10% variation, while there is little difference at solstice. Even though we consider stronger decreases in the magnetic field we still consider valid our assumption of a constant ionosphere as a first approximation. We are assuming in this case that solar radiation and solar wind will have the same characteristics during reversals as today. This detailed discussion of the impact of the magnetic field on the electron density will also be incorporated to the text.

Reference:
Cnossen, I., Richmond, A. D., Wiltberger, M., Wang, W., and Schmitt, P. ( 2011), The response of the coupled magnetosphere-ionosphere-thermosphere system to a 25% reduction in the dipole moment of the Earth's magnetic field J. Geophys. Res., 116, A12304, doi:10.1029/2011JA017063.

I recommend the authors to mainly focus on the Spitze angle, and discussed this point more in detail. The Spitze angle only depends on the magnetic field. Thus, the calculated changes of Spitze angle due to magnetic field variations are meaningful, unlike the changes of R which dominantly depend on ionosopheric density and its variability. The minor changes of R due to magnetic field variation could be shortly mentioned in discussion for completeness.

We could do this. In fact, we are preparing a paper considering the Spitze in full detail, which is a very interesting effect. But in this work we are more interested in showing the ground range dependence on the magnetic field configurations during reversal.

**Specific comments**:
a) The second part of abstract is difficult to understand without reading the article

We are working on this.

b) lines 44-46. Definitions of group and phase path using speed of light $c$ in vacuum are misleading. According to my knowledge and literature that I read the phase and group path lengths are related to distances traveled by phase and group velocities along the trajectory, respectively (group path length is simply the length of the trajectory). Anyway, I think that these terms are unnecessary for the purpose of this article and could be removed.

They will be removed.

c) paragraph on lines 54-61. I think it could be removed as similar information is better described later, e.g, in the text starting in the end of line 100 and in the following paragraph

It will be removed.

d) line 141, "*right-hand and left-hand polarization in the cases of the o- and x-mode, respectively*" That is incorrect. Ordinary mode is left-handed (L), whereas extraordinary mode is right-handed (R). The terms L-O and R-X modes are often used, instead of simply O and X modes.

We will make this correction.

e) Permittivity of vacuum is missing in equation (2)

You are right. We will correct this equation.

e) lines, 157-158, remove

These two lines explain why we use Hamilton's equations instead of simply using Snell's law. Why do you suggest removing them?

f) Equation (8), specify that $\theta$ changes along the ray path in the ionosphere. Using the initial values, it is only valid at the bottom or below the ionosphere.

You are right. And this is stated clearly in the Figure captions. The angle that presents almost no changes during the whole path is the angle between the plane containing the ray path, and the plane of the field lines, or magnetic meridian. And this last angle is included in the initial $\theta$.

g) Text related to equation (9), lines 171-179. It should be mentioned that Spitze trajectories for incidence angles between zero and critical angle $\Phi c$, $0 < \Phi < \Phi c$, are only formed for wave frequencies $f$<foF2.

You are right. We will add this comment to the revised version.

h) I think there should be sin($\Phi c$) on the left hand side of equation (9), see, e.g., Eliasson et al. (2015), J. Plasma Physics, vol. 81, 415810201, doi:10.1017/S0022377814000968 or Mjolhus (1990), Radio Sci. **25**(6), 1321–1339

You are totally right.

i) Is really the same ionospheric profile used over the same globe? If yes, it makes no sense. See also the general comment

A uniform ionosphere is used in order to obtain variations due only to magnetic field changes. It could be thought of a way to filter out the electron density effect. This is explained better in our answer to your main comment.

j) lines 268-269. The refractive index mainly depends on plasma density. Magnetic field $B$ and angle betwenn $B$ and $k$ have relatively minor effect in the ionosphere.

We will add to the revised version that "The refractive index mainly depends on plasma density"

k) line 280, least low in equatorial belt-> largest in equatorial belt
(the same in abstract)

We will do this correction in the revised version.

l) lines 325-342, this is useless here and partly out of context; without discussing the dominant effect of electron densities it makes no sense.

We wanted to highlight that the Earth magnetic field effects, despite being small, can induce errors that may be significant for certain applications.

m) lines 370-375. This text is suitable for Introduction rather than for conclusion.

We agree. We will move this paragraph to the Introduction.

Hoping to meet all your requirements,

Mariano Fagre, Bruno S. Zossi, Erdal Yigit, Hagay Amit and Ana G. Elias

---

## Referee Comment (RC2) · Anonymous Referee #2 · 22 Apr 2019

"Effects of Earth's magnetic field variation on high frequency wave propagation in the ionosphere" by Fagre et al.

The manuscript reports on HF wave propagations with respect to different conditions of internal magnetic field. Especially, the manuscript presents the propagation effects including the reflection from the lower ionosphere using ray tracing analysis. The contents may be interesting to the community for ionospheric dynamics during dipole field decrease. However, I find a serious problem that the manuscript does not deal with the effects of wave propagation by the irregularity of electron density in the ionosphere. From equations (1) to (7), I would like to point out that the effects of wave propagation by the irregularity of electron density in the ionosphere not addressing in this manuscript are dominant rather than them by geomagnetic field change focusing in this manuscript. I think that the results are not well discussed due to no discussion on the effects causing by irregularity of electron density. The author must address the advantages on this study for neglecting the wave propagation effects from the irregularity of electron density in the ionosphere. Moreover, I have also some incorrect statements in the text (see the following comments).

Comment 1 (in Section of "Theory") Most of texts and equations in this section are general talk. It looks like only equations (8) and (9) are important in this manuscript. The authors must improve making an important summary of related theory and mostly rewrite.

Comment 2 (line 242) The authors should address why the calculation took different resolution for latitudes and longitudes. I think the irregularity of electron density is smaller than the spatial resolution used in this ray tracing.

At this moment, I recommend that the manuscript requires major revisions from above serious problems related to the effects of wave propagation causing by irregularity of electron density.

---

## Author Comment (AC2) · 23 Apr 2019

**Answer to Reviewer #2:**
Thank you very much for your comments and corrections.
Following are our answer (in black) to each of your comments (in red).

General comments:
The manuscript reports on HF wave propagations with respect to different conditions of internal magnetic field. Especially, the manuscript presents the propagation effects including the reflection from the lower ionosphere using ray tracing analysis. The contents may be interesting to the community for ionospheric dynamics during dipole field decrease. However, I find a serious problem that the manuscript does not deal with the effects of wave propagation by the irregularity of electron density in the ionosphere.

You are right in that we do not deal with the significant effects of electron density variations on ray tracing (neither regular nor irregular), since the main idea of this work is to isolate the main Earth's magnetic field effect, which is a long-term as well as large-scale effect.
We will include in the revised version a thorough discussion of HF ray path effects due to ionospheric irregularities, and we will also explain why we are not considering these effects.

From equations (1) to (7), I would like to point out that the effects of wave propagation by the irregularity of electron density in the ionosphere not addressing in this manuscript are dominant rather than them by geomagnetic field change focusing in this manuscript. I think that the results are not well discussed due to no discussion on the effects causing by irregularity of electron density. The author must address the advantages on this study for neglecting the wave propagation effects from the irregularity of electron density in the ionosphere.

Indeed electron density irregularities (TIDs for example, or latitudinal variation at low latitudes) have stronger effects on HF ray paths than the Earth's magnetic field, even considering field reversal scenarios. However, the idea of the present work is to analyze the magnetic field effect, which is important for three main reasons:
1. from a theoretical point of view.
2. electron density is characterized by large amplitude high frequency variations (days, annual, decadal), whereas the magnetic field is characterized by low amplitude lower frequency variations. Our study reveals the ground range variability over a timescale of several thousand years of a reversal.
3. The Spitze angle depends only on the internal magnetic field morphology, not on the electron density.

As you suggest, and as mentioned in our answer to your first comment, we will add to the text a thorough comparison between the effects of electron density irregularities and magnetic field morphology on the ground range.

Comment 1 (in Section of "Theory") Most of texts and equations in this section are general talk. It looks like only equations (8) and (9) are important in this manuscript. The authors must improve making an important summary of related theory and mostly rewrite.

You are right. We could cite the works were the ionosphere refractive index is explained. We decided to include equations (1) to (7) thinking on the community that studies paleomagnetism,

who probably is not familiar with these concepts. But if you insist, we can delete them and just cite some papers.

Comment 2 (line 242) The authors should address why the calculation took different resolution for latitudes and longitudes. I think the irregularity of electron density is smaller than the spatial resolution used in this ray tracing.

The resolution of R space-variation is 5° in latitude and 10° in longitude. This was chosen according to the space-scale of the Earth's magnetic field variation, for which the chosen resolution is more than enough, even under a field reversal with a dominant multipolar configuration. The values were chosen in order to have well defined patterns with the least calculations. We set higher resolution in latitude because the seasonal variability of the atmosphere is more pronounced in latitude than in longitude. We tested a higher resolution, like 5°x5° and the results are almost identical. However, each ray-tracing performed at each grid point is totally different. Hamilton's equations given in Equations (15) to (20) are integrated using a Fortran subroutine which computes the numerical solutions of the system of six simultaneous first-order differential equations over a specified interval with given initial conditions applying the Adams-Moulton procedure. This is a multi-step method that requires a self-starter, in this case a fourth order Runge-Kutta method, together with a predictor given by Adams-Bashforth formula. With this method we get steps of much less than a kilometer. So, if we would analyze a TIDs effect on a ray path, it is very well resolved with our ray tracing code.

Hoping to meet all your requirements,

Mariano Fagre, Bruno S. Zossi, Erdal Yigit, Hagay Amit and Ana G. Elias